# Validation Study of the Richards-Campbell Sleep Questionnaire in Patients with Acute Stroke

**DOI:** 10.3390/jpm12091473

**Published:** 2022-09-08

**Authors:** Eleonora Rollo, Giacomo Della Marca, Irene Scala, Cristina Buccarella, Tommaso Rozera, Catello Vollono, Giovanni Frisullo, Aldobrando Broccolini, Valerio Brunetti

**Affiliations:** 1Department of Neurosciences, Università Cattolica del Sacro Cuore, 00168 Rome, Italy; 2Dipartimento Scienze dell’Invecchiamento, Neurologiche, Ortopediche e della Testa-Collo, IRCCS Fondazione Policlinico Universitario A. Gemelli, 00168 Rome, Italy

**Keywords:** sleep, stroke, questionnaire, screening, Richards-Campbell Sleep Questionnaire, polysomnography, stroke unit

## Abstract

Sleep disorders are frequent in acute stroke. The Richards-Campbell Sleep Questionnaire (RCSQ) is a validated scale for the sleep assessment in intensive care unit. The aim of the present study is to validate RCSQ for use in patients with acute stroke. We performed a validation study by comparing the RCSQ with polysomnography (PSG), the standardized measure of sleep. Inclusion criteria were age ≥ 18 years and a radiologically confirmed diagnosis of stroke. Exclusion criteria were global aphasia, extreme severity of clinical conditions and inability to attend PSG. All patients underwent PSG in a stroke unit, the day after a subjective sleep assessment by means of the RCSQ. The RCSQ was compared with PSG parameters to assess the degree of concordance of the two measures. The cohort consisted of 36 patients. Mean RCSQ score was 61.5 ± 24.8. The total score of the RCSQ showed a good degree of concordance with the sleep efficiency index of PSG. Accuracy of the RCSQ was 70%, sensitivity 71% and specificity 68%. The RCSQ is a good tool for screening the sleep quality in the setting of a stroke unit. Therefore, it could be useful to select the patients who might beneficiate from an instrumental sleep evaluation.

## 1. Introduction

Sleep disorders, such as insomnia, sleep-disordered breathing (SDB), and restless legs syndrome, are frequent in the acute phase of stroke, with an estimated prevalence of over 30% [1]. Sleep disorders and stroke have a bidirectional relation, as sleep disturbances may be both a pre-existing condition and appear de novo, because of brain damage [2]. Sleep disorders in stroke survivors are associated with increased mortality, stroke recurrence, and worse neurological and cognitive recovery [1,3]. Despite this, only about 6% of patients with acute stroke undergo a formal sleep testing [4]. The reasons for the low rate of sleep disorders screening in acute stroke is partially related to the lack of awareness among stroke physicians, but the difficult access to testing may also play a role [4]. Indeed, a routinely sleep assessment by means of polysomnography (PSG) is costly, not feasible for a large-scale screening due to its need for equipment, technicians and expert interpretation. Therefore, subjective survey instruments are needed as practical tools for sleep screening in the setting of stroke units. To date, a screening questionnaire specifically developed for the assessment of sleep in patients with acute stroke is lacking. The Richards-Campbell Sleep Questionnaire (RCSQ) is a simple, five-item visual analogue scale, validated for measuring sleep quality in intensive care unit (ICU) patients [5], Table 1. In light of the difficulty to access instrumental evaluation of sleep in such clinical setting, the RCSQ was proposed as a consistent alternative for sleep assessment in ICU [6]. A recent systematic review and meta-analysis, evaluating sleep assessment tools in critically ill patients, revealed a good correlation between the RCSQ results and the equivalent sleep variables of PSG, establishing the questionnaire as a reliable tool to assess sleep disruption in ICU [7].

To date, the RCSQ has never been validated against PSG in stroke patients, nor in the specific setting of stroke units. The aim of the present study is to validate the RCSQ for use in patients with acute stroke in the setting of a stroke unit. The secondary objective is to assess accuracy, sensitivity, and specificity of the RCSQ and of the different RCSQ items, compared with the corresponding parameters provided by PSG.

## 2. Materials and Methods

This is a validation study, performed by comparing the RCSQ with the gold standard measure of sleep (PSG). At the time of the study, patients were hospitalized at the stroke unit of the Fondazione Policlinico Universitario Agostino Gemelli, Rome, Italy. The enrollment period went on from May to September, 2021. Inclusion criteria were: age ≥ 18 years; radiologically confirmed diagnosis of ischemic or hemorrhagic stroke with onset of clinical symptoms within the previous 72 h; National Institute of Health Stroke Scale (NIHSS) score ≥ 1. Exclusion criteria were: stroke mimics; global aphasia; unstable clinical conditions requiring intubation and intensive care unit treatment and any other conditions that caused the inability of the patient to undergo PSG. The study population enrolled based on these criteria is described in Table 2. Before enrollment, all patients supplied written informed consent. The research was carried out in accordance with the Helsinki Declaration and was approved by the Catholic University of Rome’s ethical committee.

### 2.1. Clinical Information Collected

The following information were gathered: demographic characteristics (age, sex); side of the lesion; neurologic deficits at the time of registration, quantified using the NIHSS score; presence of large vessel occlusion (LVO) of the intracranial circulation; type of revascularization treatment (systemic thrombolysis, thrombectomy), when performed; the possible intake of central nervous system (CNS)-acting drugs.

### 2.2. Sleep Assessment

All patients underwent overnight unattended bed-side ambulatory PSG in stroke unit. Polygraphic tracings were recorded on a digital polygraph (Morpheus, Micromed^®^ SpA, Mogliano Veneto, Treviso, Italy); sampling rate was 256 Hz, A/D conversion 16 bit. Sleep recordings lasted from 3 p.m. to 9 a.m. in the next morning; during this period, patients were allowed to maintain their preferred sleep–wake habits. The light-off time was set at 10 p.m., and the light-on time at 7 a.m., according to the care-setting of the stroke unit. PSG montage comprehended: electroencephalogram (at least three scalp electrodes for the monitoring of the sleep–wake cycle: F4, C4, and O2 or F3, C3, and O1, referred to the contralateral mastoid), 2 electrooculograms (2 electrodes, 1 cm superior and 1 cm lateral to the external cantus of the eye, respectively, with one mastoid reference electrode), submental electromyography, intercostal electromyography (two surface electrodes above the last right intercostal space), electrocardiogram (modified D2 derivation), airflow measured with nasal cannula, thoracic and abdominal effort, arterial oxygen saturation (by digital pulse oximetry), snoring (audio recording by a vibration sensor), and body position. The PSG recordings were scored by sleep expert physicians (VB, GDM), blinded to the RCSQ questionnaires. Sleep recordings were analyzed on a computer monitor, and sleep stages were visually classified according to the criteria proposed by the American Academy of Sleep Medicine Visual Scoring Task Force for adult [8].

All patients underwent a subjective sleep quality and quantity assessment by means of the RCSQ. The RCSQ is a five-item questionnaire that globally estimates sleep by measuring perceived sleep depth, sleep latency (time it takes to fall asleep), frequency of awakenings, returning to sleep, and sleep quality. Each RCSQ item is rated from 0 to 100 on a visual analog scale, with higher scores suggesting better sleep. The overall esteem of sleep is represented by the total score, which is the average of the five categories. The RCSQ is reported in Table 1. The questionnaire was administered by neurology residents and PSG technicians, blinded to the PSG, the morning after PSG recordings.

### 2.3. RCSQ and PSG Comparison

The validation study was performed comparing the scores of the questionnaires with the corresponding parameters of PSG. In particular:Total score of the scale compared with the Sleep Efficiency Index (SEI) measured on the sleep period time (SPT, defined as the total time from the sleep onset to the last awakening) of PSG.Perceived sleep depth compared with the time spent in stage N3 over the sleep period time (N3%) provided by PSG.The subjective time to fall asleep compared with sleep latency (SL) measured by PSG.The perceived time spent awake compared with wake after sleep onset (WASO) measured by PSG.Returning to sleep after an awakening compared with the mean duration of awakenings lasting longer than two minutes (AW).The perceived quality of sleep compared with Arousal Index (AI) of PSG. Arousals were scored according to the American Sleep Disorders Association criteria [9].

### 2.4. Statistical Analysis

Numerical variables are presented as mean ± standard deviation, categorical variables are presented as number (*n*) and percentage.

The validation study was performed in two steps. As first step, the degree of concordance between RCSQ and PSG parameters was calculated and expressed by means of Kendall W coefficient. The Kendall coefficients were calculated for the total score of the scale and for each item of the scale. The Kendall W coefficient of concordance is a non-parametric test used for assessing agreement among raters. Kendall W ranges from 0 (no agreement) to 1 (complete agreement). The internal consistency of the RCSQ was tested using Cronbach’s alpha coefficient for the whole sample.

Furthermore, the Bland–Altman analysis [10] was applied to evaluate the inter-method agreement between RCSQ total score and PSG SEI. The Bland–Altman technique requires plotting the difference between the scores obtained with the methods (in our study, RCSQ–PSG) against the mean of the scores (RCSQ and PSG) to determine whether there is a bias. The bias is represented by the mean difference between the scores obtained with PSG and RCSQ, and the standard deviation of the differences [10]. A negative mean difference represents an underestimation, and a positive mean difference represents an overestimation. The limits of the agreement are calculated as follows: upper limit = mean difference + 2 SD; lower limit = mean difference − 2 SD. The 95% confidence interval for the bias and for the limits of agreement was calculated as the observed value minus t standard errors to the observed values plus t standard errors.

As a second step, we calculated accuracy, sensitivity, and specificity of each item. Each item of the questionnaire was compared with the corresponding parameter of the PSG to determine accuracy, sensitivity and specificity. When compared with PSG, a True Positive (TP) indicates that RCSQ correctly identifies the measure of interest as good, True Negative (TN) indicates that RCSQ correctly identifies the measure of interest as bad, a False Positive (FP) denotes that RCSQ misidentifies the measure of interest as good, and a False Negative (FN) means that RCSQ misidentifies the measure of interest as bad. 

Accuracy is defined as (TP + TN)/(TP + TN + FN + FP) and represents the agreement rate between PSG and the specific item of the questionnaire; sensitivity is defined as TP/(TP + FN); and specificity is defined as TN/(TN + FP).

The cut-off value of each item of the RCSQ was determined by calculating the receiver operating characteristic (ROC) curve, a plot of sensitivity vs. 1–specificity [11]. The Youden index was calculated for the total score and for each item of the RCSQ to assess the validity of the cut-off values. The Youden index is computed as sensitivity plus specificity minus 1. Its value ranges from 0 through 1. A zero value indicates that the test is useless, whereas a value of 1 indicates that there are no false positives or false negatives, i.e., the test is perfect. The cut-off values of the PSG that were used in this study were established according to those commonly employed in sleep medicine practice.

All statistics were performed by means of the Statistical Package for Social Science (SPSS^®^ software, version 22 (SPSS^®^, Inc., Chicago, IL, USA).

## 3. Results

A total of 42 patients were enrolled to study. After screening for inclusion and exclusion criteria, the final cohort consisted of 36 patients (Figure 1). Reasons for exclusion were: stroke mimics (one patient); inability to undergo RCSQ due to lack of compliance (three patients); poor quality of PSG recording (two patients). The mean age was 74.1 ± 12.9 years. Mean RCSQ score was 61.5 ± 24.8. Clinical and demographic characteristics, as well as results of polysomnography of the study cohort, are described in Table 2.

The total score of the RCSQ showed a good degree of concordance with the SEI, as expressed by a Kendall W coefficient of 0.686. Each item of the questionnaire showed a low to moderate degree of concordance with the different PSG parameters (Table 3).

The internal consistency of the RCSQ was high, as expressed by Cronbach’s alpha = 0.917, revealing a good homogeneity between items.

### 3.1. Bland–Altman Analysis

In the Bland–Altman analysis, the mean (±SD) difference found between the RCSQ and polysomnography was −14 (±23.8). The 95% confidence interval of the bias is −20.71 to −7.29. The lower limit of agreement is −61.6 (95% confidence interval −73.21; −49.99); the upper limit of agreement is 33.6 (95% confidence interval 21.99; 45.21) (Figure 2).

### 3.2. Accuracy, Sensitivity, and Specificity

The optimal cut-off value for the RCSQ total score, calculated with the ROC curve, was 65. Compared to SEI measured by PSG, accuracy of the RCSQ is 70%, sensitivity 71% and specificity 68%. The same cut-off of 65 was applied to the different RCSQ items, calculated with the ROC curves. Overall, all the items showed their highest Youden index at 65, except for Question 1 (sleep depth), which highest Youden index is at 75. However, for facilitating its use in clinical practice, the cut-off of 65 was kept for all the items, because it was the most representative and was also the best cut-off of the total score. Accuracy, sensitivity and specificity of the different items of the questionnaire are reported in Table 4. The ROC curves are represented in Figure 3.

## 4. Discussion

The aim of the present study was to validate the RCSQ for use in patients with acute stroke as a measure to estimate sleep in the setting of a stroke unit. The RCSQ was previously validated against PSG to be employed in critically ill patients [5]. It consists of five self-reported simple questions assessing quality and quantity of sleep. It can be also administered by physicians or nurses [12]. Each item is scored on a visual analogue scale ranging from 0 to 100, allowing it to be employed also in such conditions where a non-verbal communication is demanded. 

Overall, in our study the RCSQ showed a good degree of concordance with the objective standardized method for assessment of sleep. Indeed, the total score of the questionnaire had a good concordance with the chosen parameter for evaluating sleep (i.e., SEI), as expressed by the value of the Kendall W coefficient. The ROC curve suggests that the RCSQ total score of 65 is the optimal cut-off value to discriminate “good” versus “bad” sleepers. For what concerns the single items of the scale, they showed a low to moderate degree of concordance with the different PSG parameters. However, when the scores of the questionnaire where dichotomized using a cut-off value calculated by means of ROC curves, they showed an overall good sensitivity and specificity, compatible with a screening instrument. The total score of the scale showed itself a good accuracy, sensitivity and specificity for detecting the perception of good sleep. The Bland–Altman analysis revealed an acceptable agreement between the two methods, showing a moderate underestimation of sleep as expressed by the negative value of the bias. This is in line with an overall tendency to underestimate sleep in patients with a poor sleep quality [13].

Sleep disruption is a frequent condition in the acute phase of stroke: sleep architecture and sleep quality are compromised after stroke, most severely in thalamic [14] and cortical [2] strokes. A recent meta-analysis of polysomnographic studies revealed that stroke patients have poorer sleep than healthy controls, as shown by lower sleep efficiency, shorter total sleep time, and more WASO [15]. In our study, the RCSQ revealed an excellent sensitivity on detecting WASO, as well as a good sensitivity in assessing sleep efficiency. The prevalence of sleep disorders in acute stroke patients varies according to the disorder, approaching 60% for a clinically relevant sleep-disordered breathing (SDB), and 30% for a severe SDB [16]. Other highly prevalent sleep disorders in the acute phase of stroke include insomnia, periodic limb movements, and restless leg syndrome, affecting 40%, 30%, and 10% of patients, respectively [17]. Additionally, hypersomnia and excessive daytime sleepiness are frequent in the first month after stroke [18]. Sleep disorders, both pre-existent or appearing de novo, are associated with worse neurological recovery and increased cardiovascular morbidity [3,19,20]. Given the high prevalence of sleep disorders in the acute phase of stroke [1], and their impact on stroke outcome [3], a formal assessment of sleep should be part of the routinely care of stroke. Nonetheless, screening for sleep disorders in stroke patients is still underpowered [4,21]. Therefore, the use of a simple questionnaire to be employed for sleep disorders screening in stroke units may be of utmost clinical usefulness.

The limitations of the present validation study are the relatively low number of patients enrolled, and the possible heterogeneity of the patients due to the different clinical characteristics of stroke. Moreover, the majority of patients at the time of registration had low NIHSS, consistent with a mild neurological impairment, in many cases due to the efficacy of the revascularization treatments performed.

The strengths are the adoption of a gold standard tool for the assessment of sleep (i.e., PSG), and the novelty of the study, which allows to validate a scale for sleep estimation specifically for stroke patients and in the setting of a stroke unit. Furthermore, the questionnaire showed a very good internal consistency, as shown by a Cronbach’s alpha > 0.9.

In conclusion, the results of the present study suggest that the RCSQ is a good and reliable tool to assess sleep and for screening the presence of sleep disturbances in the setting of a stroke unit. Hence, it could be useful to select the patients who might beneficiate from a formal sleep evaluation by means of PSG. Indeed, the RCSQ is a low-cost instrument, it is not time consuming as it takes only few minutes to be administered, and it could be self-administered in the cases where the patient is compliant. Therefore, it could be adopted in the routinely clinical practice of stroke patients, providing a general estimation of sleep, and orienting the stroke physician to more selected sleep studies.

## Figures and Tables

**Figure 1 jpm-12-01473-f001:**
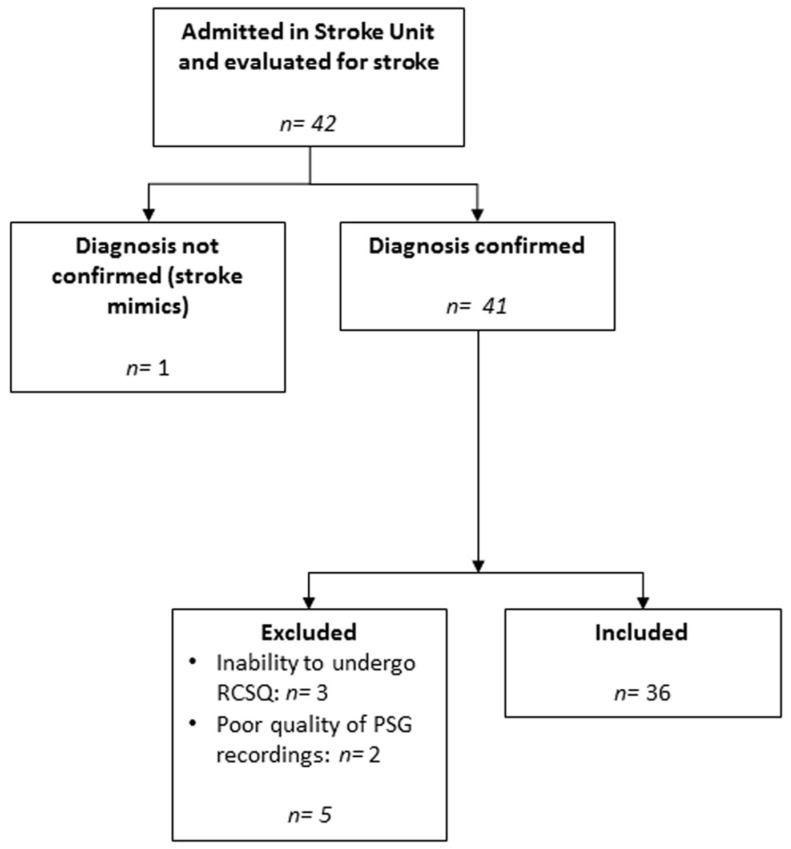
Study flowchart.

**Figure 2 jpm-12-01473-f002:**
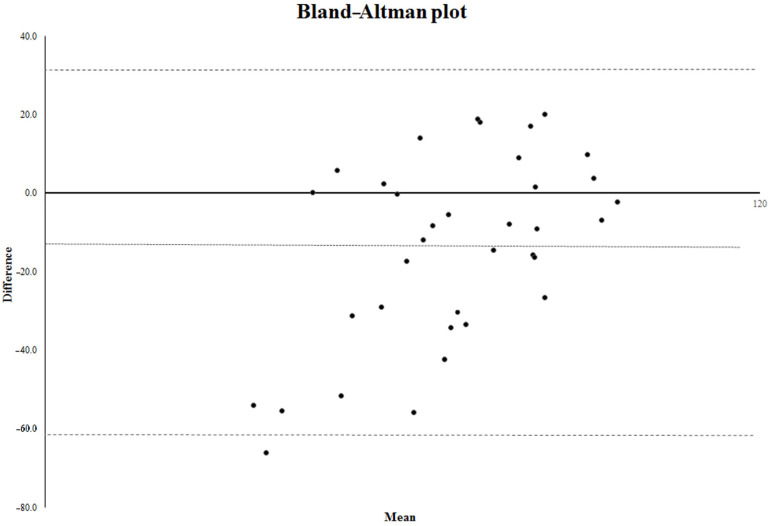
Bland–Altman analysis of agreement of the Richards-Campbell Sleep Questionnaire and the Sleep Efficiency Index measured by polysomnography. The dotted lines represent the upper and the lower limits of agreement. The spotted line represents the mean difference between the RCSQ and the PSG.

**Figure 3 jpm-12-01473-f003:**
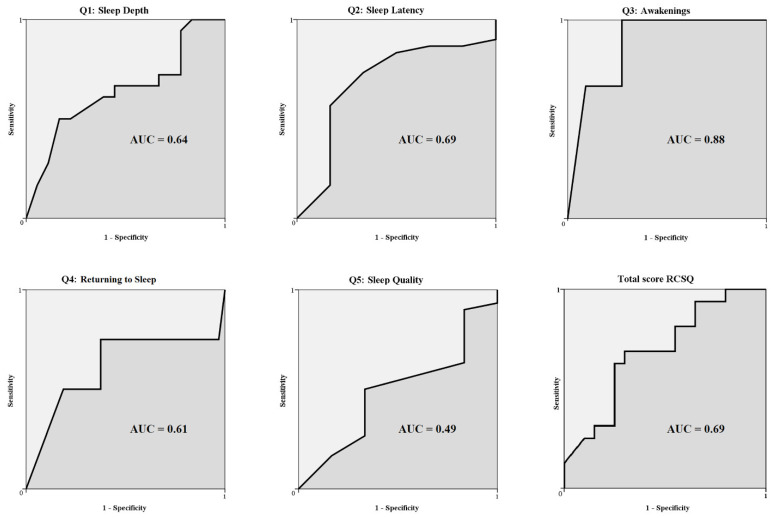
Receiver Operator Characteristic (ROC) curves for the different Richards-Campbell Sleep Questionnaire items and total score. AUC: area under curve.

**Table 1 jpm-12-01473-t001:** The Richards-Campbell Sleep Questionnaire.

Richards-Campbell Sleep Questionnaire	
Measure	Question
Q1: Sleep depth	My sleep last night was: light sleep (0) … deep sleep (100)
Q2: Sleep latency	Last night, the first time I got to sleep, I: just never could fall asleep (0) … fell asleep almost immediately (100)
Q3: Awakenings	Last night, I was: awake all night long (0) … awake very little (100)
Q4: Returning to sleep	Last night, when I woke up or was awakened, I: could not get back to sleep (0) … got back to sleep immediately (100)
Q5: Sleep quality	I would describe my sleep last night as: a bad night’s sleep (0) … a good night’s sleep (100)

**Table 2 jpm-12-01473-t002:** Demographic, clinical and polysomnographic characteristics of the study cohort. LVO, large vessel occlusion; IVT, intravenous thrombolysis; EVT, endovascular treatment; CNS, central nervous system; TST, total sleep time; SPT, sleep period time; SL, sleep latency; SEI, sleep efficiency index; WASO, wake after sleep onset; AW, mean duration of awakenings lasting more than two minutes; AI, arousal index; W, percentage of wake on sleep period time; N1, percentage of N1 stage on sleep period time; N2, percentage of N2 stage on sleep period time; N3, percentage of N3 stage on sleep period time; REM, percentage of REM stage on sleep period time; AHI-Ob, apnea-ipopnea index obstructive; AHI-C, apnea-ipopnea index central; ODI, oxygen desaturation index.

	Mean (SD)	N (%)
**Clinical data**		
Males		17 (47.2%)
Age (yr)	74.1 (12.9)	
Stroke type		
Ischemic		31 (86.1%)
Haemorragic		5 (13.9%)
side (left)		24 (66.6%)
NIHSS	3.1 (3.0)	
LVO		24 (66.6%)
IVT		11 (30.6%)
EVT		22 (61.1%)
CNS acting drugs		6 (16.6%)
**Polysomnography**		
TST (min)	366.7 (77.9)	
SPT (min)	488.0 (106.7)	
SL (min)	22.9 (34.8)	
SEI (%)	75.5 (13.9)	
WASO (min)	133.4 (74.1)	
AW (min)	32.2 (23.4)	
AI	9.4 (16.5)	
W (%)	24.9 (13.8)	
N1 (%)	8.6 (4.5)	
N2 (%)	39.2 (11.5)	
N3 (%)	15.4 (11.4)	
REM (%)	8.2 (4.9)	
AHI-Ob	28.5 (22.7)	
AHI-C	6.5 (11.6)	
ODI	32.9 (27.8)	

**Table 3 jpm-12-01473-t003:** Concordance between RCSQ total scores and RCSQ single items with the corresponding PSG parameters. SEI, sleep efficiency index; N3, duration of N3 stage on the sleep period time; SL, sleep latency; WASO, wake after sleep onset; AW, mean duration of awakenings lasting more than two minutes; AI, arousal index.

RCSQ	Mean (SD)	PSG	Mean (SD)	Concordance	Kendall W Coefficient
Total score	61.4 (24.8)	SEI (%)	75.5 (13.9)	Total score vs. SEI	0.686
Q1: sleep depth	59.8 (29.9)	N3 (%)	15.4 (11.4)	Q1 vs. N3	0.551
Q2: sleep latency	64.2 (28.0)	SL (min)	22.9 (34.8)	Q2 vs. SL	0.468
Q3: awakenings	57.5 (28.4)	WASO (min)	133.4 (74.1)	Q3 vs. WASO	0.234
Q4: returning to sleep	65.3 (29.3)	AW (min)	32.2 (23.4)	Q4 vs. AW	0.305
Q5: sleep quality	60.1 (28.7)	AI	9.4 (16.5)	Q5 vs. AI	0.453

**Table 4 jpm-12-01473-t004:** Accuracy, sensitivity and specificity of RCSQ total scores and of RCSQ single items.

	Sensitivity	Specificity	Accuracy	Cut-Off PSG	Cut-Off RCSQ
Total score RCSQ	71%	68%	70%	80	65
Q1: Sleep depth	61%	60%	61%	15	65
Q2: Sleep latency	60%	83%	64%	30	65
Q3: Awakenings	100%	58%	61%	30	65
Q4: Returning to sleep	60%	57%	58%	5	65
Q5: Sleep quality	61%	70%	64%	10	65

## Data Availability

Data are available from corresponding Authors on reasonable request.

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
