# Peer review of "Validation Study of the Richards-Campbell Sleep Questionnaire in Patients with Acute Stroke"

_jpm, 2022, doi:10.3390/jpm12091473_

Round 1

Reviewer 1 Report

This study aimed to validate the Richards-Campbell Sleep Questionnaire (RCSQ) for use in patients with acute stroke in the setting of a stroke unit. The secondary objective is to assess accuracy, sensitivity, and specificity of the RCSQ and of the different RCSQ items, compared with the corresponding parameters provided by PSG. They demonstrated that the RCSQ is a good tool for screening the sleep quality in the setting of a stroke unit. Therefore, this study suggests that the RCSQ could be useful to select the patients who might beneficiate from an instrumental sleep evaluation. The authors wrote the study results and conclusions well using appropriate statistical methods. Here are my suggestions:

1.       In Introduction, please provide a brief introduction to the Richards-Campbell Sleep Questionnaire (RCSQ) and whether there have been any previous studies examining the usefulness of RCSQ in stroke patients.

2.       How was the cut-off PSG calculated?

3.       There are some typos.

4.       What software did you use for statistical analysis?

5.       In the Bland-Altman plot (Figure 1), there is no explanation for the dotted line.

6.       ROC analysis was performed for each RCSQ item, and the same cut-off was applied to all items. Did each item show the highest Youden index at 65?

Author Response

This study aimed to validate the Richards-Campbell Sleep Questionnaire (RCSQ) for use in patients with acute stroke in the setting of a stroke unit. The secondary objective is to assess accuracy, sensitivity, and specificity of the RCSQ and of the different RCSQ items, compared with the corresponding parameters provided by PSG. They demonstrated that the RCSQ is a good tool for screening the sleep quality in the setting of a stroke unit. Therefore, this study suggests that the RCSQ could be useful to select the patients who might beneficiate from an instrumental sleep evaluation. The authors wrote the study results and conclusions well using appropriate statistical methods. Here are my suggestions:

 We thank the Reviewer for her/his appreciation and useful suggestions.

  1. In Introduction, please provide a brief introduction to the Richards-Campbell Sleep Questionnaire (RCSQ) and whether there have been any previous studies examining the usefulness of RCSQ in stroke patients.

According to the reviewer suggestions, we added a brief introduction to the RCSQ. We specified the lack of previous studies examining RCSQ in stroke patients. (Page 2, Lines 46-54).

  1. How was the cut-off PSG calculated?

The PSG cut-offs were established based on sleep medicine practice. A sentence was added in the methods section. (Page 4, Lines 159, 160)

  1. There are some typos.

The manuscript has been spell-checked.

  1. What software did you use for statistical analysis?

We specified the software for statistical analysis in the methods section. (Page4, Lines 161-162).

  1. In the Bland-Altman plot (Figure 1), there is no explanation for the dotted line.

The dotted lines represent the upper and the lower limits of agreement. The spotted line represents the mean difference between the RCSQ and the PSG. We specified the meaning of the lines in the legend to Figure 1.

  1. ROC analysis was performed for each RCSQ item, and the same cut-off was applied to all items. Did each item show the highest Youden index at 65?

Actually, most of the items showed their highest Youden index at 65, except for Question 1 (sleep depth), which highest Youden index is at 75. However, for facilitating its use in clinical practice, we decided to maintain the cut-off of 65 for all the items, because it was the most representative and was also the best cut-off of the total score. We added a sentence explaining this issue in the results section. (Page 4, Lines 184-188).

Reviewer 2 Report

Review of the manuscript -Manuscript ID: jpm-1885358

The work to evaluate the Richards-Campbell Sleep Questionnaire (RCSQ) is an interesting one. Important from the point of view of introducing/developing tools for screening sleep disorders in stroke units. As the authors write, so far the Richards-Campbell Sleep Questionnaire is used in the intensive care unit.

Suggestions for authors:

Material and method - in this section I miss a clear description of the study group. In truth, I can see Table 2 ("Table 2. Demographic, clinical and polysomnographic characteristics of the study cohort."), But it appears only in the results section with a short description of the study cohort, when the data from the table is described in sub-item "2.1. Clinical information. collected ". In my opinion, it's a bit chaotic. Please consider the suggestion.

In addition, the tables appear in a different order in the text of the manuscript. I am asking for a correction.

Item 2.2. Sleep assessment - Digital polygraph (Morpheus, Micromed®) - please add country, city, and manufacturer, if not included in the name.

The results section could include a flow diagram that well illustrates the flow of participants during the research. Please consider this possibility.

I have never come across such a small number of bibliographic items (13 items). The introduction and discussion could be extended a bit. For example, why is it so important to test sleep in post-stroke patients, do "all" centers actually do such tests, how many patients suffer from it and what are the consequences? Then the number of bibliographic items would increase.

The work requires proofreading before publication.

Author Response

The work to evaluate the Richards-Campbell Sleep Questionnaire (RCSQ) is an interesting one. Important from the point of view of introducing/developing tools for screening sleep disorders in stroke units. As the authors write, so far the Richards-Campbell Sleep Questionnaire is used in the intensive care unit.

 We thank the Reviewer for her/his appreciation and useful suggestions.

Suggestions for authors:

Material and method - in this section I miss a clear description of the study group. In truth, I can see Table 2 ("Table 2. Demographic, clinical and polysomnographic characteristics of the study cohort."), But it appears only in the results section with a short description of the study cohort, when the data from the table is described in sub-item "2.1. Clinical information. collected ". In my opinion, it's a bit chaotic. Please consider the suggestion. In addition, the tables appear in a different order in the text of the manuscript. I am asking for a correction.

According to the reviewer suggestion, in the methods section we added a reference to Table 2, where the study cohort is described (Page 2, Lines 68-69). Moreover, we revised the order of appearance of tables in the manuscript.

Item 2.2. Sleep assessment - Digital polygraph (Morpheus, Micromed®) - please add country, city, and manufacturer, if not included in the name.

The manufacturer is Micromed SpA. We added the city and the country in the text. (Page 2, Lines 82).

The results section could include a flow diagram that well illustrates the flow of participants during the research. Please consider this possibility.

We thank the reviewer for the suggestion. A figure depicting the enrolment process has been added to the manuscript (Figure 1).

I have never come across such a small number of bibliographic items (13 items). The introduction and discussion could be extended a bit. For example, why is it so important to test sleep in post-stroke patients, do "all" centers actually do such tests, how many patients suffer from it and what are the consequences? Then the number of bibliographic items would increase.

We thank the reviewer for the suggestion. The introduction and the discussion have been extended accordingly, and further bibliographic references have been added.

The work requires proofreading before publication.

The manuscript has been grammar- and spell-checked.